# Non-Coding RNA Signatures of B-Cell Acute Lymphoblastic Leukemia

**DOI:** 10.3390/ijms22052683

**Published:** 2021-03-07

**Authors:** Princess D. Rodriguez, Hana Paculova, Sophie Kogut, Jessica Heath, Hilde Schjerven, Seth Frietze

**Affiliations:** 1Department of Biomedical and Health Sciences, University of Vermont, Burlington, VT 05405, USA; princess.rodriguez@uvm.edu (P.D.R.); hana.paculova@med.uvm.edu (H.P.); sophia.kogut@uvm.edu (S.K.); 2The University of Vermont Cancer Center, University of Vermont, Burlington, VT 05405, USA; jessica.l.heath@med.uvm.edu; 3Department of Biochemistry, University of Vermont, Burlington, VT 05405, USA; 4Department of Pediatrics, University of Vermont, Burlington, VT 05405, USA; 5Department of Laboratory Medicine, University of California, San Francisco, CA 94143, USA; Hilde.Schjerven@ucsf.edu

**Keywords:** non-coding, leukemia, B-cell, RNA-sequencing, small RNA-sequencing

## Abstract

Non-coding RNAs (ncRNAs) comprise a diverse class of non-protein coding transcripts that regulate critical cellular processes associated with cancer. Advances in RNA-sequencing (RNA-Seq) have led to the characterization of non-coding RNA expression across different types of human cancers. Through comprehensive RNA-Seq profiling, a growing number of studies demonstrate that ncRNAs, including long non-coding RNA (lncRNAs) and microRNAs (miRNA), play central roles in progenitor B-cell acute lymphoblastic leukemia (B-ALL) pathogenesis. Furthermore, due to their central roles in cellular homeostasis and their potential as biomarkers, the study of ncRNAs continues to provide new insight into the molecular mechanisms of B-ALL. This article reviews the ncRNA signatures reported for all B-ALL subtypes, focusing on technological developments in transcriptome profiling and recently discovered examples of ncRNAs with biologic and therapeutic relevance in B-ALL.

## 1. Introduction

Leukemia is a cancer of developing blood cells that can occur at any age, and is the most common cancer in children, accounting for nearly one-third of all pediatric cancers [1]. The most common type of pediatric leukemia is acute lymphoblastic leukemia (ALL). ALL is characterized by the rapid growth of abnormally developing lymphoid cells within the bone marrow, which restricts the production of normal blood cells. As a result of improvements in the treatment of ALL, including the development of targeted therapies, the five-year survival rate for children with standard risk ALL is 93% [2,3,4]. However, relapsed ALL remains a major cause of cancer-related death and the prognosis for ALL declines drastically with age, where the five-year survival rates in adults 40 years and older remains just over 30% [5].

Acute lymphoblastic leukemia is classified by cell lineage. The two main forms of ALL are B lymphoblastic and T lymphoblastic ALL [6]. B lymphoblastic ALL (B-ALL) is the most common form of pediatric cancer, accounting for nearly 80% of pediatric ALL [7]. B-ALL can be further classified according to recurrent driver genomic lesions including chromosomal aneuploidy, rearrangements that result in fusions that deregulate proto-oncogenes or transcription factors, and point mutations [4]. Accordingly, cytogenetic and molecular analysis is important for the clinical management of B-ALL, providing a basis for risk stratification, and in some cases treatment with targeted therapies [8]. For example, among the aneuploidy groups, one subset is termed high hyperdiploidy (HdH; containing 51–65 chromosomes) and is found in one-third of children with ALL [9]. Similarly, the reciprocal translocation t(12;21)(p13;q22) generating the fusion gene *ETV6/RUNX1* (also known as TEL/AML1) is also common, accounting for nearly 25% of pediatric B-ALL cases [8]. Clinically, the prognosis is favorable for both HdH and *ETV6/RUNX1* subtypes with five-year overall survival rates near 80% [9,10]. In contrast, in high-risk subtypes such as BCR-ABL1-positive (also known as Philadelphia chromosome-positive, Ph+) B-ALL the incidence increases with age and is associated with poorer prognosis [8]. Specifically, patients with Ph+ makes up approximately 5% of B-ALL cases in children and account for 40–50% of B-ALL cases in adults [5]. Outcomes for Ph+ B-ALL have improved with the administration of imatinib, a tyrosine kinase inhibitor (TKI), however resistance to these treatments can occur [11,12]. Despite the utility of prognostic biomarkers, unanswered questions remain regarding the relationship of genomic alterations to leukemogenesis, chemotherapy resistance, and clinical outcome. As new molecular signatures of B-ALL emerge, the possibility of improved prognostication and treatment stratification may reduce treatment failure and relapse associated with B-ALL.

Global gene expression profiling has provided considerable new insight into the molecular basis of B-ALL, including the identification of distinct high-risk subgroups [13,14]. Moreover, transcriptome-wide profiling by RNA-Seq has identified non-coding RNAs (ncRNAs) as central players in cancer progression [15]. Diverse classes of ncRNAs including small and long non-coding RNAs (lncRNAs) have been defined as key regulatory factors that may have significant roles in determining cancer phenotypes [16]. A growing number of studies indicate that these two major types of ncRNAs may play critical roles in the pathogenesis and progression of B-ALL and may serve as prognostic biomarkers or provide opportunities as new therapeutic targets. This review describes ncRNAs and their functions, focusing on technological developments in transcriptome profiling and recently discovered examples of ncRNAs in B-ALL with biologic and therapeutic relevance.

## 2. A Compendium of Human Non-Coding RNA Genes

A remarkable number of genes in the human genome are transcribed into protein-coding and various non-coding RNA species (Figure 1). Comprehensive and current gene annotations provided by the GENCODE [17] and RefSeq projects [18] are indispensable resources for the investigation of the coding and non-coding RNA landscape of the human genome [19], where non-coding genes represent nearly half of all annotated genes. 

The non-coding genes can be grouped into two broad categories according to the nucleotide (nt) length of the RNA transcripts: long non-coding RNA (lncRNA; >200 nt) and small non-coding RNA (<200 nt). The number of non-coding RNA genes annotated in the current GENCODE annotation (version 36) includes 17,958 lncRNA genes and over 7569 small RNA genes, comprising approximately 30% and 13% of all the annotated human genes in this reference. Due to the continual emergence of new data, as well as the improvements and increased application of long-read sequencing technologies, these gene annotation resources are continually updated, providing a more complete annotation of ncRNA genes [20]. Additionally, many resources exist to curate and systematically classify ncRNAs including the HUGO, NONCODE and LNCipedia databases [21,22,23]. 

### 2.1. miRNAs Are Regulatory Small Non-Coding RNAs

In general, ncRNAs have been shown to function in numerous physiological and developmental processes. The small RNA genes are comprised of several different biotypes including microRNA, snRNA, snoRNA, rRNA, tRNA, vaultRNA and Y-RNA. Many of these small ncRNAs with abundant and ubiquitous expression across cells participate in general housekeeping functions, including mRNA splicing and translation (i.e., rRNAs, tRNAs, snoRNAs, etc.). Additionally, some small RNAs (i.e., miRNAs, Y-RNAs, etc.) can be released from the cell and are detectable in biofluids such as plasma, serum, and bronchoalveolar lavage fluid [24]. 

Studies have found that many small ncRNAs are considered dynamic regulatory RNA molecules with functional roles in post-transcriptional gene regulation. In particular, microRNAs (miRNAs) are well known to play central regulatory roles through post-transcriptional gene regulation via direct binding to mRNAs (reviewed in [25]). It is predicted that the majority (> 60%) of mRNAs can be bound by miRNAs [26] and accordingly, miRNAs have been widely connected to diverse human diseases, including cancer and B-ALL, as reviewed below [27]. MiRNA genes are transcribed as long primary transcripts (pri-miRNAs) and are sequentially processed into precursor miRNAs (pre-miRNAs) and then into mature miRNAs, approximately 22 nucleotides in length. As ribonucleoprotein complexes mature, miRNAs bind specific mRNAs that are complementary to the miRNA sequence, which results in target mRNA degradation or translational repression (for a review, see [28]). The miRNA:mRNA interaction is subject to various regulatory steps, including the subcellular location of miRNAs, the expression levels of miRNAs and various target mRNAs, as well as the expression of other RNA transcripts that alter miRNA function. 

### 2.2. Regulatory Long Non-Coding RNA (lncRNA)

Similar to protein-coding mRNAs, many lncRNA genes are transcribed by RNA polymerase II, then capped, spliced and polyadenylated [29]. As a group, lncRNA genes are generally poorly conserved across mammals and exhibit low expression levels and/or cell or tissue-specific expression patterns [30]. In an effort to classify lncRNAs, lncRNA genes are grouped based on their position relative to protein-coding genes, including: (i) intergenic lncRNA, (ii) antisense lncRNA, (iii) divergent lncRNA (bidirectional promoter), (iv) intronic lncRNA, and (v) overlapping lncRNA [31]. In comparison to miRNAs, whose functions have been extensively characterized, much less is understood about lncRNA genes as functional studies have only been performed for a limited number of lncRNAs [21]. 

Despite there being only a small fraction of lncRNA genes described in the literature, current evidence indicates that lncRNAs play diverse biological roles. Depending on their specific interactions with DNA, RNA and proteins, lncRNAs have been reported to modulate chromatin structure, control the assembly and function of various nuclear complexes, regulate the translation of mRNAs, and restrict various cellular signaling pathways (reviewed in [32]). The different modes of lncRNA action can be grouped into five broad categories, as follows. First, lncRNAs contribute in the *cis*-regulation of neighboring genes. This can occur through chromatin modulation whereby lncRNAs interact with DNA to form RNA-DNA hybrids such as R-loops that recruit transcription factors [33] or chromatin modifiers [34,35,36] to modulate gene transcription or to directly induce transcription [37]. Second, lncRNAs participate in the *trans*-regulation of a gene locus (or many loci) distal from the lncRNA gene through similar mechanisms involving chromatin modification [38,39]. Third, lncRNAs act as structural modules that mediate protein and RNA complexes (ribonucleoproteins or RNPs). LncRNA-containing RNP complexes have been implicated in a wide variety of cellular processes ranging from transcription, pre-mRNA processing, chromatin modification, and cellular signaling (reviewed in [40]). Fourth, lncRNAs can function as molecular “decoys” or “mimics” to sequester proteins or small RNAs from their targets. For example, an abundant lncRNA called NORAD (for “noncoding RNA activated by DNA damage”) binds to the RNA binding proteins PUM1 and PUM2 to insulate them from the mRNAs to which they bind, which include genes involved in chromosome segregation during cell division [41,42]. Finally, numerous lncRNAs have been described as “sponges” that sequester miRNAs from their target mRNAs. The lncRNAs which function in this manner are commonly referred to as competing endogenous RNAs (ceRNAs). These sponges or ceRNAs harbor a range of miRNA response elements (MREs) that are complimentary to miRNAs. These lncRNA-miRNA interactions have been described (e.g., [43,44,45]).

## 3. Profiling Non-Coding RNAs

RNA-Seq is a powerful tool for the analysis of ncRNAs and is the process by which the RNA in a sample is first converted to a cDNA library for DNA sequencing via Next Generation Sequencing (NGS). There are many advantages to RNA-Seq over other profiling assays such as microarrays. Namely, microarrays require predefined transcript sequences for probe hybridization, whereas RNA-Seq facilitates the single base-level unbiased analysis of transcriptomes permitting the discovery of novel ncRNA sequences. Despite being the overall preferred method for ncRNA profiling, RNA-Seq approaches also have certain disadvantages, including the cost per sequencing run and the processing steps required to convert an RNA sample into a cDNA library for sequencing. In addition, small RNAs (~20 to 50 nucleotides in size) are not efficiently captured in cDNA libraries, thus a separate library protocol is required to generate libraries for small RNA profiling. For this reason, to most comprehensively capture a complete ncRNA transcriptome for a given sample, both small RNA-Seq and RNA-Seq library protocols should be performed on a given sample (Figure 2). 

Small RNA-sequencing (smRNA-Seq) is a sensitive technique used to detect the expression of miRNAs and other small RNA species with large dynamic range. Typically, the smRNA library preparation process involves the ligation of adapters to RNA prior to reverse transcription and PCR amplification. A primary technical challenge for this method is that different small RNAs may be either over- or underrepresented in the cDNA sequencing library [46,47]. Underrepresented small RNAs that are lowly expressed transcripts may not be detected. Furthermore, the 3′ terminal nucleotide of various small RNAs may carry a 2′-O-methyl (2′ OMe) modification which can strongly reduce the efficiency of 3′ adapter ligation, thus making library preparation particularly challenging for small RNAs containing such modifications [48]. Moreover, the RNA ligases used in the adapter ligation reactions exhibit both sequence and structural preferences for different RNAs contributing to variation in the composition of different small RNAs in the cDNA library [48]. To overcome adapter ligation issues, improved protocols are available [49]. 

Most lncRNAs are routinely analyzed via standard RNA-Seq libraries (versus the smRNA-seq protocol; Figure 2). However, it is necessary to enrich informative RNA (lncRNA, mRNA, etc.) and deplete abundant RNA including ribosomal RNA (rRNA). Typically, rRNA and other abundant transcripts such as globin found in blood samples are depleted from total RNA prior to library construction. The depletion process relies on single-strand oligonucleotide probes that are complimentary to the target RNAs to be depleted (i.e., rRNA or globin) to form RNA:DNA hybrids that can be degraded with RNAse H, and then DNAse to remove the probes [50,51]. Alternatively, polyadenylated (poly-A+) transcripts may be enriched by using oligo-dT probes coupled to magnetic beads during RNA-Seq library preparation; this is a standard method in many commercial RNA-seq kits/services. However, of the greater than 15,500 human lncRNAs transcripts catalogued, only 39% contain at least one of the six most common poly(A) motifs [52]. Therefore, an oligo-dT enrichment method may not capture as many lncRNAs as rRNA-depletion methods and should be considered when profiling lncRNA expression [53]. 

It is also common to find genomic loci where both strands encode distinct genes, called anti-sense genes. In the human genome, there are an estimated 11,000 overlapping genes that can be transcribed from the opposite strands (about 17% of all genes) [54]. For lncRNAs specifically, it has been reported that about one-third of human lncRNA loci are anti-sense [52]. As lncRNAs can be transcribed from either strand of the genome, it is important to perform a stranded RNA-Seq protocol that retains the strand information [54]. Strand-specific RNA-Seq library protocols facilitate the alignment and quantification of transcripts derived from the opposite strands at the same position of the genome. Of note, newer library preparation methods are being developed that simultaneously capture and evaluate both small- and long- ncRNAs. Giraldez et al. proposed an alteration to the smRNA-Seq protocol termed “phospho-RNA-seq” [55]. In this adaptation, a T4 polynucleotide kinase step is added to phosphorylate 5′ and dephosphorylate 3′ ends of RNA, thus permitting adapter ligation and extracellular RNA fragment detection by small RNA-Seq. 

## 4. Long Non-Coding RNAs in Progenitor B-Cell Acute Lymphoblastic Leukemia

Several large-scale transcriptomic studies have described global gene expression patterns of normal and malignant human hematopoietic cells (e.g., reviewed in [56,57]). The RNA landscape of the normal human hematopoietic hierarchy generated by RNA-Seq and smRNA-Seq profiling methods from purified HSCs and their various differentiated progenies is now available [58]. Resources such as BloodSpot, Haemosphere, and other publicly available databases contain valuable datasets for exploring the transcriptional networks that underlie specific blood lineages and their associated hematological malignancies [59,60,61]. For example, hundreds of lncRNAs exhibit cell-type/stage-specific expression patterns across B-cell development and maturation in humans and mice, whereby distinctive subsets of progenitor B-cells can be distinguished by lncRNA expression patterns [62,63,64,65,66]. By mapping patient samples onto this landscape and other normal expression profiles, signatures of upregulated lncRNAs in patient samples as well as subtype-specific lncRNAs in B-ALL are emerging. Although the number of profiling studies is limited, they clearly indicate that lncRNA expression patterns are deregulated in B-ALL [67,68,69]. An assembled list of lncRNAs reported in B-ALL is provided in Table 1. The candidate lncRNAs identified in these studies provide a strong basis for further studies aiming to identify their function or assess their potential clinical value.

While B-ALL transcriptome studies can be limited by the inclusion of a restricted number of subtypes, small biological sample sizes, or the use of microarrays that prevent the identification of novel transcripts, they continue to provide a valuable overview of B-ALL specific lncRNAs. Among the first studies to perform lncRNA expression profiling on different B-ALL subtypes, Dinesh Rao and colleagues found that lncRNA expression patterns predict the cytogenetic profile of B-ALL for three common B-ALL subtypes (*ETV6-RUNX1* t(12;21)(p13;q22), *TCF3-PBX1* t(1;19) and *MLL-AF4* t(4;11)) [68]. Termed “B-ALL–associated long RNAs” or BALR, this study identified the differentially expressed lncRNAs BALR-1, BALR-2, BALR-6, and LINC00958. In particular, the lncRNA BALR-2 correlated with a poor patient response to prednisone and a worse overall survival. In support of a functional role of this lncRNA in B-ALL, this same study showed that depletion of BALR-2 resulted in an increase in apoptosis of B-ALL cell lines alone, and in combination with glucocorticoids. Conversely, overexpression of BALR-2 resulted in increased cell proliferation and glucocorticoid resistance. Overall, this study suggests that BALR-2 may function to promote B-ALL cell survival and may be a determinant of glucocorticoid response.

A more recent study by Lajoie et al. performed RNA-Seq analysis in 56 patient samples to identify the expression of lncRNAs across several subtypes of B-ALL [70]. When compared to CD10+CD19+ pre-B-cells isolated from human cord blood, BALR-1 and LINC00958 were also found to have increased expression in *ETV6-RUNX1* subtypes. Interestingly, BALR-2 expression was found to be elevated in *MLL*-rearranged patients and in patients harboring either t(4;11) or the t(9;11) translocations. To investigate a functional role for these newly identified B-ALL lncRNAs, Ouimet et al. used siRNA to deplete 5 candidate lncRNAs identified in the aforementioned study [70,80]. In particular, depletion of the lncRNA RP11-137H2.4 in human NALM-6 leukemia cells inhibited cell proliferation and migration, in addition to, restored glucocorticoid sensitivity in resistant cells. Further to this study, Gioia et al. functionally characterized three lncRNAs downregulated in B-ALL (RP-11-624C23.1, RP11-203E8, and RP11-446E9), by restoring their expression in REH cells and in the non-leukemic cell line GM12878 for comparison [83]. The examined lncRNAs exhibited tumor suppressor properties, promoted apoptosis in response to DNA damaging agents, and upon restoration in leukemic cells exhibited reduced proliferation and migration when compared to non-leukemic cells. The lncRNA LINC00958 identified by Lajoie et al. [70] as previously reported to function as a ceRNA in pancreatic cancer cells, where it interacts with miR-330-5p to regulate PAX8 levels [72]. Further analysis needs to be performed to determine if LINC00958 and other lncRNAs deregulated B-ALL function as ceRNAs. 

The expression of lncRNAs may also be directly influenced by proteins arising from translocation events occurring in B-ALL. A study by Ghazavi et al. used microarrays to profile specimens from 64 B-ALL samples that consisted of *ETV6-RUNX1*, *TCF3-PBX1*, high hyperdiploid, and normal karyotype genetic subtypes [79]. After integrating patient lncRNA expression data with RNA-Seq results generated from a panel of human B-ALL leukemic cell lines, this study reported the unique lncRNA expression profile of 16 lncRNAs exclusively associated with the presence of the ETV6-RUNX1 fusion protein. Of these, lncRNA SARDH-1 (also known as lncRNA DBH-AS1) was found to be downregulated in *ETV6-RUNX1*-positive cells. LncRNA SARDH-1/DBH-AS1 has been previously studied and found to promote cell proliferation and cell survival in multiple different cancer models [84,85,86,87,88]. This study also compared the altered transcriptional responses after silencing the *ETV6-RUNX1* fusion transcript in the REH cell-line. They found 134 lncRNAs to be deregulated (41 up- and 93 down-regulated). Of these, four lncRNAs (lnc-NKX2-3-1, lnc-TIMM21-5, lnc-ASTN1-1 and lnc-RTN4R-1) overlapped with the previously determined lncRNA signature found in primary *ETV6-RUNX1* samples. Furthermore, functional analysis using the lncRNA and mRNA expression profiles revealed mRNA processing and vincristine resistance to be among the top correlated gene sets for all four lncRNAs mentioned above. In a separate study performed by Cuadros et al., the transcriptional responses in *ETV6-RUNX1*-positive patient samples were compared to *ETV6-RUNX*-negative samples [75]. This study found 117 differentially expressed lncRNAs between the two sample types. Of these, the highest expressed lncRNA was TCL6 and was predicted to coregulate the mRNA *TCL1B* in *ETV6-RUNX1*-positive B-ALL. Importantly, this study indicated that low TCL6 levels may also be associated with poor disease-free survival, even within *ETV6-RUNX1* B-ALL, a favorable sub-group of pediatric B-ALL. Overall, these studies suggest that lncRNA expression analysis could complement current cytogenetic and molecular analyses applied in the routine diagnosis of B-ALL to further refine current standards of risk stratification.

The expression of lncRNAs as indicators of treatment response or relapse have gained significant attention in cancer research (reviewed in [89,90]). Understanding if lncRNA expression could serve as useful indicators of disease prognosis in B-ALL has also been investigated. Indeed, transcriptome analysis performed on bone marrow samples collected from *DUX4*, Ph-like, or near-haploid and high hyperdiploid (NH-HeH) B-ALL patients collected at initial diagnosis (ID) and relapse (REL) revealed 1235 subtype-specific lncRNAs [91]. After performing differential analysis between samples taken at ID to REL, 947 total lncRNAs were deregulated in the three subtypes, with the majority being downregulated at relapse [91]. When comparing ID to REL samples within each subtype studied, 570, 113, and 248 lncRNAs were differential in *DUX4*, Ph-like, and in NH-HeH samples respectively. Other studies have shown that LINC00152 and LINC01013 were among the most differentially expressed genes in patients with early relapse of disease compared to healthy controls [76]. Specifically, higher expression of LINC00152 in children with B-ALL was associated with a higher risk of early relapse; conversely, lower expression of LINC01013 was associated with early relapse. The lncRNA LINC00152 is also known as CYTOR (long non-coding RNA cytoskeleton regulator RNA) and has been previously described as an oncogenic lncRNA in many different types of cancer (e.g., [92,93,94,95,96,97]) and may serve as a potential biomarker for cancer detection in both solid tumor tissue and plasma [98,99]. Overall, these reports indicate that lncRNA expression can serve as useful biomarkers in B-ALL.

### Notable Case: LncRNA GAS5 and Its Connection to Glucocorticoid Resistance

Drug resistance, whether intrinsic or acquired, is also a major problem in the management of ALL (reviewed in [89]). Glucocorticoids like dexamethasone and prednisone are critical in the treatment of B-ALL where they exhibit cytotoxic activity against hematological cells through the activation of apoptosis [100]. When given in combination with chemotherapy during induction therapy, 85% of ALL patients from age 16 to 80 years achieved complete remission [101]. However, a failed response to initial glucocorticoid therapy is associated with unfavorable prognosis and poor outcome [102,103]. Additionally, prolonged use can lead to the emergence of glucocorticoid resistance. Studies have found that lncRNA expression may be functionally associated with drug resistance in acute leukemias [68,80,89,104]. For example, the lncRNA growth arrest-specific transcript 5 (GAS5) is downregulated in many cancers and may be used as a potential marker of treatment response in remission induction therapy for children with ALL [81,82,105]. 

The GAS5 gene contains 12 exons and multiple spliced and polyadenylated isoforms have been reported [106]. The primary transcripts are ~600 nucleotides in length. The accumulation of GAS5 transcripts is impacted by various cellular stressors, including serum or nutrient deprivation as well as various drug treatments [107]. In cases such as these, where there is an inhibition of translation of GAS5 transcripts, cell proliferation and survival are negatively regulated (Figure 3A). There is also early evidence suggesting that modulation of GAS5 expression could restore glucocorticoid sensitivity in healthy blood mononuclear cells [81,108]. 

The primary effects of glucocorticoids occur via activation of the glucocorticoid receptor (GR) (encoded by the *NR3C1* gene), a ligand-activated transcription factor belonging to the superfamily of nuclear receptors. The GR-binding site was mapped to the 3′ terminus of GAS5 (bases 546–566), a region that forms a putative stem-loop structure (Figure 3B). This region forms an explicit RNA–protein interaction domain that binds directly to the helix 1 of the GR DNA binding domain to block DNA-dependent glucocorticoid signaling [109,110]. Thus, in steroid-sensitive cancer cells, such as leukemia cells, the GR binding motif is responsible for GAS5 effects on cell growth [109]. However, this may not be true in other cell types where proliferation is not necessarily dependent on GR signaling. Indeed, Frank et al., has used structure-function analysis to examine the other regions of GAS5 lncRNA that regulate cell survival [107]. Namely, functional studies performed by this group in Jurkat and CEM-C7 T-ALL cell lines show that GAS5 also possesses a 5′ region that confers the ability of GAS5 to reduce cell survival, as well as a core region that is required for mediating the effects of mTOR inhibition. Thus, GAS5 possesses independent structural modules that function to regulate growth in various cellular conditions through distinctive mechanisms.

Functionally, the GAS5 lncRNA interacts with the DNA binding domain of ligand-activated GR and negatively regulates GR transcriptional activity by inhibiting binding of GRs to glucocorticoid response elements (GREs) (Figure 3C) [110]. GAS5 may also function in leukemia cell drug responsiveness through other additional mechanisms not dependent on GR. For example, GAS5 has been shown to regulate mTOR/AKT pathways in cancers [111,112,113]. Alterations in the mTOR pathway have been implicated in leukemogenesis [114]. Furthermore, GAS5 may mediate regulatory interactions of miRNAs and target mRNAs involved in drug responses (reviewed in [115]), including miR-222 in leukemia cells [116]. Together these studies highlight GAS5 as an important lncRNA whose structure influences glucocorticoid drug responsiveness and essential pathways important for cell survival and growth.

## 5. MicroRNA and Progenitor B-Cell Acute Lymphoblastic Leukemia

MiRNAs play key biological roles during B-cell development and their expression patterns are carefully regulated across distinct developmental stages [117,118]. As we continue to discover and study miRNAs, those previously found to have important roles in lymphopoiesis may also contribute to B-cell malignancies. There are currently 1917 miRNAs registered in the *Homo sapiens* miRNA registry (miRBase version 22) [119]. Of these, the LeukmiR database predicts 861 miRNAs are associated with ALL [120]. One example is miR-150, which is a critical regulator of B-cell development and is primarily expressed in the lymph nodes and spleen [121]. In B-cells, miR-150 expression progressively increases during development where it is expressed at low levels in progenitor cells and then becomes abundantly expressed in immature and mature B-cell subsets [121]. Interestingly, ectopic expression of miR-150 has been shown to inhibit the pro-B to pre-B transition and can modulate B-cell receptor signaling [122,123]. In B-ALL, microarray analysis has shown that miR-150 is lowly expressed in pediatric patients compared to healthy controls [124]. Further, this study found that miR-150 expression changed among prednisone response groups and was down-regulated in relapsed patients compared to controls. In an effort to understand the role of miRNAs in B-ALL, several groups have performed gene-expression studies in leukemic samples with the goal of identifying miRNAs that could be used as diagnostic and prognostic markers, as well as markers of resistance to existing therapies [125,126]. An assembled list of miRNAs reported in B-ALL is provided and discussed below (Table 2).

Many types of leukemias exist and it has been shown that miRNA expression can distinguish between the major leukemic types and subtypes. For example, miRNA profiling performed in 72 patient samples identified 27 miRNAs that were differentially expressed between ALL and acute myeloid leukemia (AML) samples [127]. Among these, 4 miRNAs (miR-128a, miR-128b, let-7b and miR-223) were found to be sufficient to discriminate between the two leukemic types. A later study performed by Zhang et al. also confirmed that miR-128a and miR128b in addition to five others (miR-213, miR-210, miR-130b, miR-146a, and miR-34a) were characteristic of ALL and differentially expressed when compared to healthy controls [124]. Further, this study identified two prognostic miRNA expression signatures that could be used to differentiate between good or poor prednisone response or discriminate between relapsed and control cases in ALL. Similarly, other studies have found that T-cell lineage ALL has a distinct pattern of miRNA expression when compared to B-cell lineage ALL. Almeida et al. performed miRNA transcriptome sequencing on samples collected from 4 T-ALL and 4 B-ALL patients. They reported 16 deregulated miRNAs (6 up- and 10 down-regulated) which could be used to distinguish between the two groups [128]. Gene expression studies performed by other laboratories confirmed that miR-425-5p and miR-126 were indeed downregulated in T-ALL compared to B-ALL [129,130]. Interestingly, miR-126 has been recently found to be highly expressed in hematopoietic stem cells; additionally, overexpression of miR-126 in B-cell progenitors resulted in leukemic initiation in a murine model [131].

MiR-125b has been found to be deregulated in many different cancer types [132]. In B-ALL, miR-125b was found to be highly expressed in patients harboring the translocation t(11;14)(q24;q32), which involves the immunoglobulin heavy chain locus, and in patients with *ETV6-RUNX1* fusions [130,133]. Studies performed in mice found that when miR-125b is co-expressed with the *BCR-ABL* fusion gene, this resulted in an accelerated development of leukemia [134]. Furthermore in B-ALL patients, high expression of miR-125b along with miR-99a and miR-100 were shown to serve as markers of resistance to vincristine [130,135]. Overall, these results suggest that miR-125b may play a role in ALL initiation and can be used a marker of drug resistance. 

MiRNA signatures can also be predictive of disease outcome and relapse. Upon performing miRNA profiling with paired diagnosis-relapse samples, Han et al. identified miR-708, miR-223, and miR-27a as differentially expressed in relapsed childhood ALL [136]. Similarly, another group identified miR151-5p, miR-451, and miR-1290 as relapse markers when correlating miRNA profiles of patient samples at diagnosis with clinical outcome [137]. Together, these studies indicate that miRNA gene expression correlates with leukemic type and can potentially be used in a clinical setting to assess responsiveness to chemotherapeutic drugs as well as risk of relapse.

## 6. Other Classes of Non-Coding RNAs and Their Role in Progenitor B-Cell Acute Lymphoblastic Leukemia

Thus far, this review has focused on describing miRNAs and lncRNAs identified as having significance in B-ALL. However, other classes of ncRNAs are emerging as biologically significant factors in cancer. In particular, dysregulation of circular RNAs (termed circRNAs) may contribute to the development and progression of cancer [reviewed in [138]. CircRNAs are a large family of ncRNA molecules that have been recently established to function in a variety of central biological processes including transcription, translation, mRNA splicing and RNA decay [139,140]. CircRNAs are single-stranded, covalently closed RNA molecules that are derived from pre-mRNAs through a process called back-splicing [141]. Relative to their linear counterparts, circRNAs are stable and can be detected in exosomes, saliva, and plasma [142,143,144], suggesting that circRNAs are potential clinical biomarkers for disease progression and prognosis [reviewed in [145]. CircRNA expression can be detected with RNA-Seq, and their expression patterns have been found to be unique to various human tissues and blood cells [139]. For example, in the hematopoietic system the majority (~80%) of circRNAs detected in monocytes and B, T-cells isolated from healthy donors were found in all three cell types, while 10% of circRNAs expressed were specific to the lymphocyte population [146]. 

CircRNA expression was found to be deregulated in B-ALL cells. After comparing circRNA expression in normal B-cells to patient derived xenograft B-ALL cells, seven circRNAs were identified as being significantly different in B-ALL [146]. Importantly, the expression of certain circRNAs, such as circAFF2, appeared subtype-specific and follow-up studies should be performed to understand if circAFF2 could be used as a biomarker for the TCF3-PBX1 subtype of B-ALL. In other studies, performed by Huang et al., the circRNA circAF4 was identified as having an oncogenic role in *MLL-*rearranged leukemias [147]. More on the role of circRNAs in hematopoiesis and in hematological malignancies can be found [reviewed in [148,149,150]. 

Although not extensively studied in B-ALL, expression profiling performed in other hematological diseases such as multiple myeloma [151], chronic lymphocytic leukemia [152], and in AML [153], has revealed that small nucleolar RNAs (snoRNAs) could also serve as useful biomarkers in hematological malignancies. SnoRNAs are a class of small noncoding RNAs that are generally less than 300 nucleotides in length and have been shown to regulate mRNA splicing, serve as endogenous sponges, and function as guide RNAs during post-transcriptional modification of target RNAs [154,155]. Microarray studies performed by Valleron et al. using AML and ALL samples found that many snoRNAs were downregulated compared to normal myeloid or lymphocyte cells respectively [156]. Additional follow-up studies would contribute to the understanding if unique snoRNA signatures exists for B-ALL.

## 7. Emerging Perspectives

Traditionally, risk stratification has been based on clinical factors such as age, white blood cell count, and response to chemotherapy. The identification of recurrent genetic abnormalities has helped refine risk stratification and guide treatment decisions. In the past decade, major advancements have been made to comprehensively profile samples from cohorts of patients with leukemia. In particular, a resource available to researchers granting access to genomic sequencing data collected from thousands of leukemic patients is the National Cancer Institute’s Therapeutically Applicable Research to Generate Effective Treatments (TARGET). The TARGET project has facilitated a multi-omic analysis of patients with ALL and provides an exceptional resource for the exploration of gene expression data in cancer research. This shared resource provides multiple genomic datasets including RNA-Seq and small RNA-Seq files collected from patient cohorts with diverse leukemic subtypes. Moreover, integrative genomic analysis has led to the identification of 23 unique subtypes of B-ALL after analyzing whole-transcriptome profile and cytogenetic analysis from 1988 patients [157]. Further, subtypes that had not been previously described were further characterized with whole-genome sequencing (WGS), whole-exome sequencing (WES), and single-nucleotide polymorphism (SNP) array analysis [157]. Importantly, some patients that had been formally categorized in the “B-other ALL” group were newly classified into a defined subtype based on mutational status and distinct gene expression signature [157].

Data sharing is the future of science and the utilization of these large patient datasets could potentially identify new determinants of drug resistance, biomarkers, and refinements in patient risk stratification with the incorporation of altered non-coding transcripts. Aberrant ncRNA expression has been shown to serve as prognostic indicators to predict treatment response or disease relapse for various malignancies [158,159,160,161]. Additionally, due to high cell-type specific expression patterns, ncRNA signatures may act as non-invasive biomarkers for diagnosis and prediction of treatment outcomes [16]. For example, 95% of pancreatic patients overexpress the lncRNA PCA3 (Prostate Cancer gene 3), also known as DD3, which has restricted expression to the prostate tissue [162]. Due to its high expression in cancer cells, PCA3 is detectable in the urine and upon quantification can be used to generate a PCA3 score to predict biopsy outcome [163,164,165]. In 2012, PCA3 was FDA-approved as the only urinary biomarker for prostate cancer [166,167,168]. Other lncRNAs such as HOTAIR, MALAT1, and GAS5 also show promise as cancer biomarkers [169,170,171]. In leukemia, higher GAS5 expression was shown to correlate with poor overall survival [172]. Functional studies should be performed to understand the role of GAS5 in B-ALL. As the non-coding genome is not completely annotated, the biological functions of many miRNAs and lncRNAs are unknown and more experimental work is required to clarify clinical relevance. In the future, we can envision comprehensive studies, integrating coding and non-coding transcriptomic data from large patient cohorts to identify new determinants of drug resistance and therapeutic targets. 

## Figures and Tables

**Figure 1 ijms-22-02683-f001:**
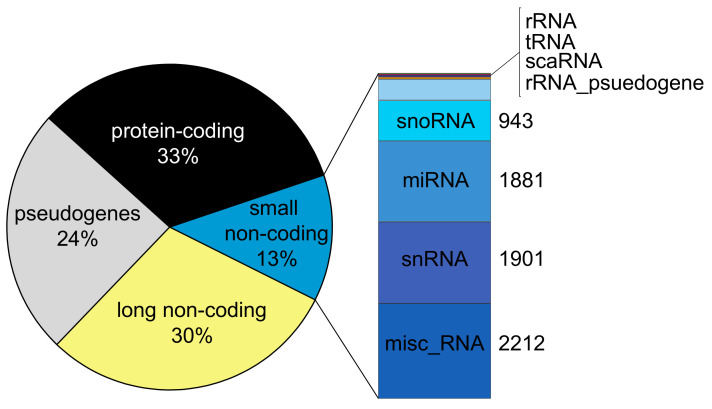
Diagram showing the major classes of genes contained within the human genome based on GENCODE annotation version 36 (https://www.gencodegenes.org/human/ (accessed on 6 March 2021)). The small RNA genes are composed of a variety of different biotype annotations. Shown in the stacked barplot are the small non-coding gene types, along with numbers of genes per category from the HGNC project (https://www.genenames.org/download/statistics-and-files/ (accessed on 6 March 2021)). (Abbreviations: rRNA, ribosomal RNA; tRNA, transfer RNA; scaRNA, Small Cajal body-specific RNA; snoRNA, small nucleolar RNA; miRNA, microRNA; snRNA, small nuclear RNA; misc_RNA, miscellaneous RNA).

**Figure 2 ijms-22-02683-f002:**
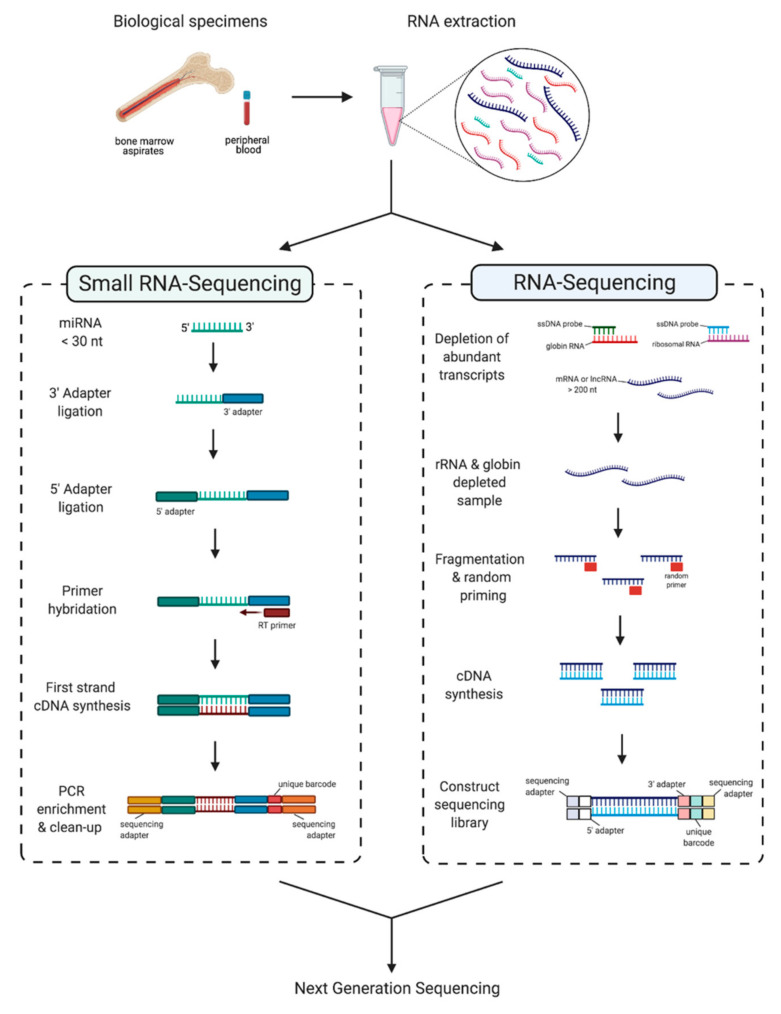
Profiling non-coding RNAs with transcriptome sequencing in B-ALL. In the study of B-ALL, input sample types will vary and can include biological specimens such as the bone marrow, peripheral blood, cord blood, or plasma. RNA is harvested from cells and then assessed for quality. Extracted total RNA from one sample can then be split and used for downstream library construction to profile both small- and long- ncRNAs. Small RNA-sequencing library protocols incorporate additional steps that are required to sequence small RNA species, typically <30 nucleotides. Long non-coding RNAs, as well as mRNAs, are detected with the conventional RNA-Sequencing library preparation protocol. Created with BioRender.com.

**Figure 3 ijms-22-02683-f003:**
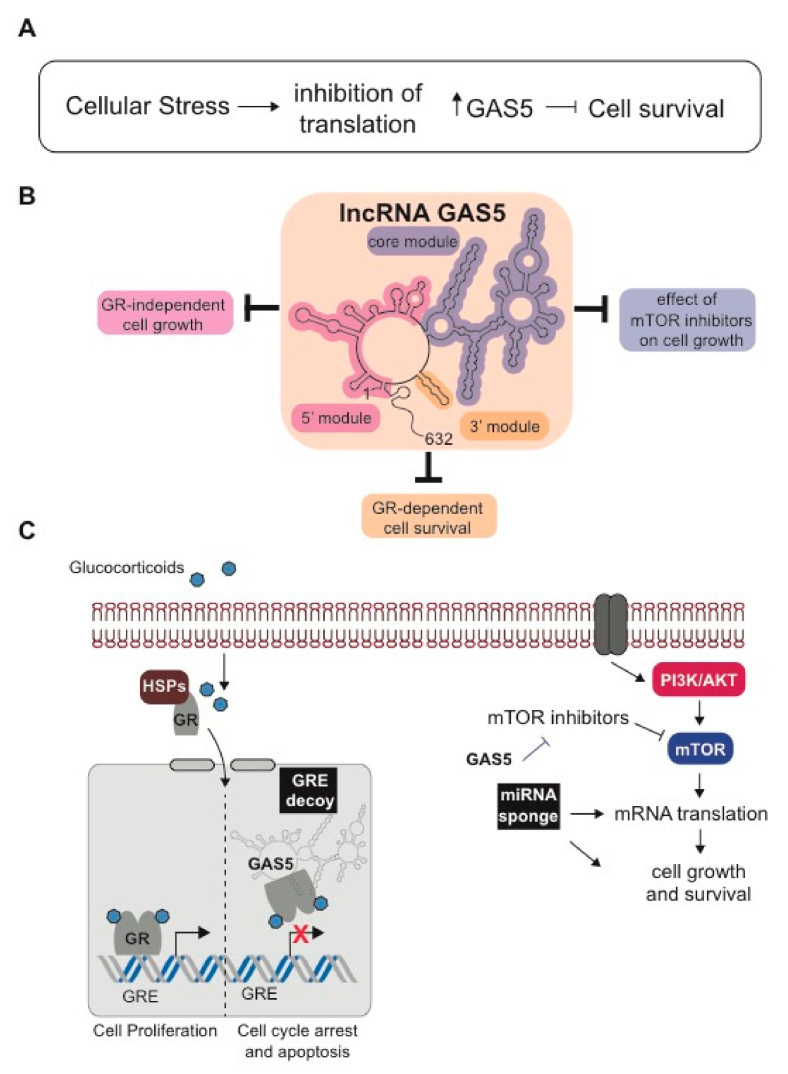
GAS5 lncRNA regulates cellular stress responses through multiple mechanisms. (**A**) Various cellular stressors, including serum or nutrient deprivation as well as various drug treatments, result in the inhibition of translation and the accumulation of GAS5 transcripts, whereby GAS5 negatively regulates cell proliferation and survival. (**B**) The GAS5 RNA molecule comprises three separate regions that mediate the effect of GAS5 on cell growth through distinctive mechanisms. These regions are divided into 3 modules: a 5′ unstructured region, a highly structured core region, and a 3′ terminal glucocorticoid receptor (GR) binding region. Functional analysis indicates that the 5′ module mediates the effect of GAS5 on basal cell survival and reducing the rate of cell cycle, whereas the core module is required for mediating the effects of mTOR inhibitors. (**C**) The mechanisms by which GAS5 regulate cell proliferation in GR-dependent (left) and GR-independent means (right) are shown. For the GR-dependent process, GAS5 functions as a glucocorticoid response element (GRE) decoy effectively sequestering GR from GRE elements in DNA and suppressing GR-dependent gene expression and cellular proliferation. For the GR-independent growth control, GAS5 can function as a miRNA sponge to regulate signaling pathways. Additionally, inhibition of the mTOR pathway downstream of PI3K/AKT signaling depends on GAS5.

**Table 1 ijms-22-02683-t001:** Examples of lncRNAs with significance in B-cell acute lymphoblastic leukemia.

LncRNA	Neighboring Genes	B-ALL Subtype Expression	Clinical or Functional Implications	References
BALR-1	*C14orf132*	upregulated in *ETV6-RUNX1* and High hyperdiploid subtypes	Unknown	[68,70]
BALR-2/CDK6-AS1	*CDK6*	*ETV6-RUNX1*, TCF3-PBX1 and MLL-rearranged subtypes	High expression correlated with poor overall survival and reduced response to prednisone treatment	[68,70]
BALR-6	*SATB1, TBC1D5*	highest expression in *MLL*-rearranged subtypes	Promotes cell survival in vitro	[68,71]
LINC00958	*TEAD1, RASSF10*	upregulated in *ETV6-RUNX1*	miRNA sponge	[68,70,72,73,74]
TCL6	*TCL1B*	*ETV6-RUNX1*	Low expression associated with poor disease-free survival	[75]
AL133346.1	*CCN2*	unknown	High AL133346.1/CCN2 expression associated with greater disease-free survival	[69]
LINC00152/CYTOR	intergenic; resides in a cluster of lncRNAs on 2p11.2	unknown	High expression associated with risk of early relapse	[76]
LINC01013	intergenic; resides in a cluster of lncRNAs on 6q23.2	unknown	Low expression associated with risk of early relapse	[76]
LAMP5-AS1	*LAMP5*	*MLL*-rearranged	High expression associated with reduced disease-free survival	[67,77]
CASC15/LINC00340	*SOX4*	*ETV6-RUNX1*	Regulates expression of SOX4	[68,78]
DBH-AS1	*DBH*	unknown	Promotes cell survival through activation of MAPK signaling	[79]
lnc-NKX2-3-1	*NKX2-3*	upregulated in *ETV6-RUNX1*	Unknown	[79]
lnc-TIMM21-5	*NETO1*	upregulated in *ETV6-RUNX1*	Unknown	[79]
lnc-ASTN1-1	*ASTN1*	upregulated in *ETV6-RUNX1*	Unknown	[79]
lnc-RTN4R-1	*RTN4R, CCDC188*	upregulated in *ETV6-RUNX1*	Unknown	[79]
RP11-137H2.4/lnc-DYDC1-1	*TSPAN14, SH2D4B*	upregulated in B-ALL compared to control pre-B cells isolated from human cord blood	Associated with cell survival and glucocorticoid resistance in vitro	[70,80]
GAS5		High hyperdiploid and TCF3-PBX1	Associated with glucocorticoid treatment sensitivity	[81,82]

**Table 2 ijms-22-02683-t002:** Examples of microRNAs with significance in B-cell acute lymphoblastic leukemia.

miRNA	B-ALL Expression	Cohort Description	Clinical or Functional Implications	References
miR-125b	Upregulated in TEL-AML1-positive compared to non-TEL-AML1 precursor B-ALL	Mononuclear cells were isolated from BM and PB from 81 ALL patients, of which 70 were of the B-ALL subtype and 11 were T-ALL. 17 control samples were also included^a^	Highly expressed (along with miR-100and miR-99a) in patients resistant to vincristine^a,d^	[130]^a^, [135]^b^, [133]^c^, [134]^d^
	Upregulated in patients with t(11;14)(q24;q32) compared to B-ALL patients without t(11;14)	Total RNA was extracted from samples taken from 2 patients with t(11;14)(q24;q32) translocations and 28 B-ALL patients without t(11;14) for qPCR^b^ Mononuclear cells were isolated from BM and PB from patients diagnosed with *ETV6*-*RUNX1*, *TCF3* (*E2A*)-rearrangement, *MLL*-rearrangement or *BCR*-*ABL1.* Validation experiments was performed on Reh cells^d^	When co-expressed with *BCR-ABL*, was shown to accelerate the development of leukemia in mice^c^	
miR-425-5p	Upregulated in B-ALL compared to T-ALL patients	See above^a^ BM or PB was obtained from 8 patients with ALL. Of these, 4 patients had T-ALL and 4 had B-ALL^e^ BM or PB was obtained from 20 patients with ALL and analyzed by miRNA array. Of these, 4 had T-ALL and 16 had B-ALL. In the B-ALL cohort, 4 patients had a *BCR/ABL* rearrangement, 3 had an *E2A/PBX1*, 3 had an *MLL/AF4* rearrangement, and 6 patients had no molecular abnormalities^f^	Unknown	[130]^a^, [128]^e^ [129]^f^
miR-126	Upregulated in TEL-AML1-positive compared to non-TEL-AML1 precursor B-ALL	See above^a,f^	Higher expression correlated with chemotherapy resistance^a^	[130]^a^, [129]^f^ [131]^g^, [124]^h^
	Higher expression in BCR/ABLcohort compared to T-ALL patients	BM aspirates were collected from 17 B-ALL samples; 16 samples were further studied. 11 were of the Ph+ B-ALL subtype and 5 were of the B-ALL ‘other’ subtype^g^	In xenotransplant murine model, knockdown induced apoptosis of B-ALL blast cells^g^	
	Upregulated in B-ALL compared to healthy controls	BM samples from 43 patients were analyzed by microarray. These included 18 ALL, 18 AML, and 7 normal samples. Among the ALL samples, 17 were of the B-cell lineage^h^		
miR-34a	Upregulated in B-ALL compared to healthy controls	See above^h^	Unknown	[124]^h^
miR-130b	Upregulated in B-ALL compared to healthy controls	See above^h^	Unknown	[124]^h^
miR-146a	Upregulated in B-ALL compared to healthy controls	See above^h^	Unknown	[124]^h^
miR-213	Upregulated in B-ALL compared to healthy controls	See above^h^	Highly expressed in high-risk and intermediate risk groups; not abnormally expressed in standard-risk group.	[124]^h^
miR-210	Upregulated in B-ALL compared to healthy controls	See above^h^	Highly expressed in high-risk and intermediate risk groups; not abnormally expressed in standard-risk group.	[124]^h^
miR-128a	Upregulated in B-ALL compared to AML samples and when compared to healthy controls.	See above^h^ BM samples were collected from 58 patients for miRNA expression analysis. Of these, 11 were B-ALL and 47 were AML. All B-ALL samples had *MLL*-rearrangements. 14 cell lines were also included (7 ALL and 7 AML). In addition, 3 BM samples were collected from healthy controls^i^	Highly expressed in ALL; can be used in miRNA expression signature to discriminate ALL from AML	[124]^h^, [127]^i^
miR-128b	Upregulated in B-ALL compared to AML samples and when compared to healthy controls.	See above^h,i^	Highly expressed in ALL; can be used in miRNA expression signature to discriminate ALL from AML	[124]^h^, [127]^i^
miR-708	Upregulated at relapse compared to complete remission in matched-paired ALL samples	Matched-paired samples were collected from 18 ALL patients at diagnosis and at relapse or complete remission for microarray studies. Of these, 11 patients had B-ALL. 5 healthy control samples were also included^j^	Higher expression correlated with higher relapse free survival in newly diagnosed ALL patients	[136]^j^
miR-1290	Upregulated in ALL patients with adverse clinical parameters compared to those with good clinical parameters	BM samples from 48 patients were analyzed by microarray of which 35 were of the B-cell lineage and 13 were of the T-cell lineage. 32 of the B-ALL samples from the initial cohort, in addition to, 106 added B-ALL samples (n=132) were used for confirmation studies^k^	High expression was associated with increased risk of relapse	[137]^k^
miR-151-5p	Downregulated in ALL patients with adverse clinical parameters compared to those with good clinical parameters	See above^k^	Low expression was associated with increased risk of relapse	[137]^k^
miR-451	Downregulated in ALL patients with adverse clinical parameters compared to those with good clinical parameters	See above^k^	Low expression was associated with increased risk of relapse	[137]^k^
miR-150	Downregulated in relapsed B-ALL patients compared to complete remission	See above^h^	Low expression was associated with poorer response to prednisone and is a part of a miRNA signature used to discriminate between relapse and complete remission	[124]^h^
Let-7b	Downregulated in *MLL*-rearranged compared to compared to *MLL*-negative patients	See above^a,i^	Target analysis identified *c-MYC* and *RAS* as downstream targets of the let-7 family. mRNA levels of *c-MYC* and *RAS* were upregulated in *MLL*-rearranged ALL compared to non-MLL B-ALL patients^a^	[130]^a^, [127]^i^
	Downregulated in B-ALL compared to AML samples		Lowly expressed in ALL; can be used in miRNA expression signature to discriminate ALL from AML^i^	
miR-223	Downregulated in B-ALL compared to AML samples	See above ^i, j^	Lowly expressed in ALL; can be used in miRNA expression signature to discriminate ALL from AML^i^	[127]^i^, [136]^j^
	Downregulated at relapse compared to complete remission in matched-paired ALL samples		Higher expression correlated with higher relapse free survival in newly diagnosed ALL patients	
miR-27a	Downregulated at relapse compared to complete remission in matched-paired ALL samples	See above ^j^	Higher expression correlated with higher relapse free survival in newly diagnosed ALL patients	[136]^j^

Abbreviations: AML: acute myeloid leukemia, ALL: acute lymphoblastic leukemia, BM: bone marrow, PB: peripheral blood, B-ALL: B lymphoblastic ALL, T-ALL: T lymphoblastic ALL. a–k Correspond to references listed: a doi:10.3324/haematol.2010.026138, b doi:10.1016/j.leukres.2013.06.027, c doi:10.1038/leu.2010.93, d doi:10.1073/pnas.1016611107, e doi:10.1002/hon.2567, f doi:10.1002/gcc.20709, g doi:10.1016/j.ccell.2016.05.007, h doi:10.1371/journal.pone.0007826, i doi:10.1073/pnas.0709313104, j doi:10.1093/hmg/ddr428, k doi:10.1002/gcc.22334.

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
