# Peer review of "Non-Coding RNA Signatures of B-Cell Acute Lymphoblastic Leukemia"

_ijms, 2021, doi:10.3390/ijms22052683_

Round 1

Reviewer 1 Report

The topic is thouroughly described and the review easy to read and understand. From my side, no changes need to be made.

The authors have in this review summarized the current literature regarding non-coding RNAs in B-Cell acute lymphoblastic leukemia. As the authors highlight in their work, in recent years non-coding RNAs in disease have gained increasing interest and particularly, as explained in the review, they play a prominent role in B-ALL. The review is very comprehensive, well explained and covers all possible areas related to the topic. Additionally, the review provides summary tables on non-coding RNAs and their significance in B-ALL as well as a very comprehensive table of microRNAs in B-ALL, this makes it very easy for the reader to extract important information without having to read the text. Furthermore, to my knowledge, it is the first review focusing on non-coding RNAs in B-ALL, thus it is an important and novel work.

Author Response

Thank you for your endorsement. 

Reviewer 2 Report

The authors clearly describe ncRNAs in B-ALL with an original approach. However, in my opinion, some minor points should be addressed for publication:

Consistently with the text, the authors should indicate the incidence of Ph+ B-ALL in the introduction.

An explanation of abbreviations for various types of ncRNAs represented in Figure 1 should be provided in the legend and possibly in paragraph 2.

Sentences related to ceRNAs are repeated in paragraphs 2.1 and 2.2, I suggest to mention them only in the lncRNA paragraph.

I think there is a typo in the text: Linc00958 in Table 1 is mentioned as Linc0098 in the text (lines 255 and 265). Moreover, it is reported as miRNA sponge in the table: there are some evidences of this or other lncRNAs as sponges in B-ALL?

The role of circRNAs in B-ALL should be also described (‘Circular RNAs in Blood Malignancies https://doi.org/10.3389/fmolb.2020.00109’ as review). Furthermore, since they have been mentioned in figure 1, also studies on the role of snoRNAs in B-ALL should be analyzed. I suggest to add one/two new paragraphs for these other classes of ncRNAs.

Therapeutic perspectives on the use of ncRNAs in B-ALL treatment should be mentioned in the last paragraph. Refer to ‘Non-Coding RNAs: The “Dark Side Matter” of the CLL Universe’ Pharmaceuticals https://doi.org/10.3390/ph14020168 as review.

Author Response

The authors clearly describe ncRNAs in B-ALL with an original approach. However, in my opinion, some minor points should be addressed for publication:

Consistently with the text, the authors should indicate the incidence of Ph+ B-ALL in the introduction.

Thank you for bringing up this point. We have modified the text in lines 50-54 to include both the percentage and incidence of Ph+ B-ALL in children and adults.

An explanation of abbreviations for various types of ncRNAs represented in Figure 1 should be provided in the legend and possibly in paragraph 2.

Thank you for this suggestion. We have modified Figure 1 to include an explanation of abbreviations of the various types of ncRNAs. In addition, we have included a comprehensive list of abbreviations following the conclusions sections. 

Sentences related to ceRNAs are repeated in paragraphs 2.1 and 2.2, I suggest to mention them only in the lncRNA paragraph.

We agree with this suggestion. We have removed ceRNA example from paragraph 2.1.

I think there is a typo in the text: Linc00958 in Table 1 is mentioned as Linc0098 in the text (lines 255 and 265). Moreover, it is reported as miRNA sponge in the table: there are some evidences of this or other lncRNAs as sponges in B-ALL?

Thank you for the correction. It is indeed a typo that has been fixed to Linc00958 in the manuscript. In addition, lines 280-284 have been added to expand upon Linc00958 as a miRNA sponge in pancreatic cancers.

The role of circRNAs in B-ALL should be also described (‘Circular RNAs in Blood Malignancies https://doi.org/10.3389/fmolb.2020.00109’ as review). Furthermore, since they have been mentioned in figure 1, also studies on the role of snoRNAs in B-ALL should be analyzed. I suggest to add one/two new paragraphs for these other classes of ncRNAs.

Thank you for this suggestion. We have added the section “6. Other classes of non-coding RNAs and their role in in progenitor B-cell Acute Lymphoblastic Leukemia” to describe both circRNAs and snoRNAs in B-ALL.

Therapeutic perspectives on the use of ncRNAs in B-ALL treatment should be mentioned in the last paragraph. Refer to ‘Non-Coding RNAs: The “Dark Side Matter” of the CLL Universe’ Pharmaceuticals https://doi.org/10.3390/ph14020168 as review.

Thank you for this suggestion. See lines 518-530.